# Dynamic of Glucose Homeostasis in Virtual Patients: A Comparison between Different Behaviors

**DOI:** 10.3390/ijerph19020716

**Published:** 2022-01-09

**Authors:** Alexis Alonso-Bastida, Manuel Adam-Medina, Rubén Posada-Gómez, Dolores Azucena Salazar-Piña, Gloria-Lilia Osorio-Gordillo, Luis Gerardo Vela-Valdés

**Affiliations:** 1Electronic Engineering Department, TecNM/CENIDET, Cuernavaca 62490, Morelos, Mexico; manuel.am@cenidet.tecnm.mx (M.A.-M.); gloria.og@cenidet.tecnm.mx (G.-L.O.-G.); luis.vv@cenidet.tecnm.mx (L.G.V.-V.); 2TecNM/CRODE, Orizaba 94380, Veracruz, Mexico; rposadag@ito-depi.edu.mx; 3Faculty of Nutrition, UAEM, Cuernavaca 62350, Morelos, Mexico; azucena.salazar@uaem.mx

**Keywords:** daily activities, extended Bergman minimal model, glucose homeostasis, Monte Carlo approach

## Abstract

This work presents a mathematical model of homeostasis dynamics in healthy individuals, focusing on the generation of conductive data on glucose homeostasis throughout the day under dietary and physical activity factors. Two case studies on glucose dynamics for populations under conditions of physical activity and sedentary lifestyle were developed. For this purpose, two types of virtual populations were generated, the first population was developed according to the data of a total of 89 physical persons between 20 and 75 years old and the second was developed using the Monte Carlo approach, obtaining a total of 200 virtual patients. In both populations, each participant was classified as an active or sedentary person depending on the physical activity performed. The results obtained demonstrate the capacity of virtual populations in the generation of in-silico approximations similar to those obtained from in-vivo studies. Obtaining information that is only achievable through specific in-vivo experiments. Being a tool that generates information for the approach of alternatives in the prevention of the development of type 2 Diabetes.

## 1. Introduction

Day by day it is more common to see in society cases of chronic degenerative diseases derived from metabolic diseases such as diabetes mellitus, this is largely a product of poor diet promoted by the large amount of food products with high amounts of glucose and sedentary habits that most people present throughout the day. In Mexico, the National Health and Nutrition Survey reported that 8,542,718 people over 20 years of age had a previous medical diagnosis of type 2 diabetes mellitus (T2DM) [1]. For this reason, it is necessary to have a better understanding of the effects that factors such as diet and physical activity have on glucose homeostasis.

According to Williams et al. [2], in 2019 Mexico ranked sixth in the world in terms of economic investment for the treatment of diabetes (17.0 billion dollars invested), for this reason it is important to seek alternatives that improve the management and prevention of the disease in the country, through projections that provide more information on sectors vulnerable to suffering from this condition. Environmental factors [3], family history [4], age [5], overweight [6], polycystic ovary syndrome [7], hypertension [8], abnormal cholesterol levels [9] and triglyceride levels [10], poor diet [4], as well as sedentary lifestyles [11], are important parameters in the development of metabolic diseases, for this reason, it is necessary to pay more attention to these factors to improve disease prevention strategies, considering changes in lifestyles, diet, regular physical activity and maintaining a healthy body weight as alternatives for the prevention and reduction of these ailments as well as cardiovascular problems [4,11].

From the mathematical point of view, the problem of metabolic diseases has been approached on the basis of in-silico experiments, which in conjunction with mathematical models generate numerical approximations of the behavior of glucose homeostasis when food intake occurs. An example of this are precursor studies such as Dalla Man et al. [12] who propose a mathematical model of glucose homeostasis under conditions where the patient may be healthy or have T2DM; Kovatchev et al. [13] and Man et al. [14] based on mathematical models develop virtual patients (children, adolescents and adults) with type 1 diabetes mellitus (T1DM) for the development of applications in-silico related to glycemic control.

Based on the mathematical models of glucose homeostasis, a new alternative of studies focused on the analysis of virtual populations has emerged, where virtual patients (VP) are developed with particular characteristics for the study to be performed. An example of this is the work proposed by Orozco-López et al. [15], where physically healthy VP are developed with T1DM based on minimum order models, Visentin et al. [16] on the other hand generates VP with T1DM and Visentin et al. [17] creates VP with T2DM according to maximum order models, which are some examples of the current panorama in the research of VP development. This type of research seeks to ensure the feasibility of in-vivo experiments from those previously developed. In several works, in-silico studies based on a population of analyzed VP are proposed, in Garcia-Tirado et al. [18] predictive control techniques of multistage models for artificial pancreas (AP) systems based on 100 VP with T1DM are developed, in Rahmanian et al. [19], a controller is developed to regulate blood glucose levels in patients with T1DM, where they relied on in-silico experimentation. In Nath et al. [20] according to 100 VP with T1DM nonlinear observers based on glucose control are designed. Similar situations are presented in Bhattacharjee et al. [21], Lee et al. [22], Rashid et al. [23] and Toffanin et al. [24] where in-silico experimentation in AP systems plays an important role to obtain an approximation of the results that can be obtained in in-vivo experimentation.

Physical activity is a factor that improves glucose homeostasis as well as body weight control [25]. Several papers present mathematical models describing variations in the dynamics of glucose homeostasis according to physical activity for experimental development in-silico. In Alkhateeb et al. [26], studies on minimum order mathematical models for the development of VP that consider variations of the glycemic profile in patients with T1DM under food intakes as well as physical activity are presented; a simulator for the development of VP with T1DM under conditions of physical activity, insulin administration and food intakes is shown in Kartono et al. [27], Resalat et al. [28], Garcia-Tirado et al. [18] and Moser et al. [29] research concerning the effects of physical activity on glucose homeostasis. Therefore, it is possible to observe that most of the attention of metabolism studies from the mathematical point of view are focused on conditions of diabetes mellitus, especially the case of type 1, being this a reduced sector of the population with metabolic diseases.

Healthy people constitute the majority of the world’s population, however, due to society’s unhealthy lifestyles, this leads to an increased risk of various metabolic diseases. The main objective of this work is to generate a mathematical modeling alternative capable of generating conductive data on the dynamics of glucose homeostasis throughout the day under factors such as diet and physical activity. Obtaining graphical and numerical approximations of virtual populations on various scenarios of glucose variation, being a support in decision making on the management of metabolic diseases.

## 2. Materials and Methods

A mathematical model is a mathematical tool focused on the generation of numerical approximations for different dynamics in physical processes. In glucose homeostasis these models are used as a tool to develop control solutions for insulin management in insulin-dependent patients and approximations for experimental protocol results in population studies. The representation of the proposed mathematical model is presented below.

### 2.1. Mathematical Representation

To simulate the different dynamics of glucose homeostasis in healthy people under the effect of factors such as diet and physical activity. In this work, an extended representation of Bergman’s minimal model is proposed, combining the models proposed in Orozco-López et al. [15] and Roy and Parker [30], in addition to proposing mathematical representations to simulate the effects of diet and physical activity as the main drivers of variations in glucose homeostasis.

#### 2.1.1. Glucose Homeostasis Representation

The extended representation of the Bergman minimal model starts from the variables of plasma glucose concentration, insulin concentration and the effect of glucose lowering by insulin derived from Orozco-López et al. [15] and Roy and Parker [30] and described in Equations (Equation 1)–(4).
(1)$$\begin{array}{cc} \; & \frac{dI(t)}{dt}=-\eta \left(I\left(t\right)-I_{b}\right)-\gamma \left(G\left(t\right)-h\right)t-I_{e}\left(t\right); \end{array}$$
(2)$$\begin{array}{cc} \; & \frac{dX(t)}{dt}=-p_{2}X\left(t\right)+p_{3}\left(I\left(t\right)-I_{b}\right); \end{array}$$
(3)$$\begin{array}{cc} \; & \frac{dG(t)}{dt}=-p_{1}\left(G\left(t\right)-G_{b}\right)-X\left(t\right)G\left(t\right)+\ldots \end{array}$$
(4)$$\begin{array}{cc} \; & \ldots +\frac{W}{Vol_{G}}\left[G_{prod}\left(t\right)-G_{gly}\left(t\right)\right]-\frac{W}{Vol_{G}}G_{up}\left(t\right)+d\left(t\right). \end{array}$$
where *I* is the plasma insulin concentration (μU/mL), *X* is the insulin effect on the glucose concentration reduction (1/min), *G* represents the blood glucose concentration (mg/dL) and the subscript *b* represent the baseline states of model.

#### 2.1.2. Effect of Physical Activity on the Dynamics of Glucose Homeostasis

Based on the following mathematical representations proposed by Roy and Parker [30], the glucose homeostasis relationships are described, according to physical activity effects carried out.
(5)$$\begin{array}{cc} \; & \frac{dG_{prod}\left(t\right)}{dt}=a_{1}PVO_{2}^{max}\left(t\right)-a_{2}G_{prod}\left(t\right); \end{array}$$
(6)$$\begin{array}{cc} \; & \frac{dG_{up}\left(t\right)}{dt}=a_{3}PVO_{2}^{max}\left(t\right)-a_{4}G_{up}\left(t\right); \end{array}$$
(7)$$\begin{array}{cc} \; & \frac{dI_{e}\left(t\right)}{dt}=a_{5}PVO_{2}^{max}\left(t\right)-a_{6}I_{e}\left(t\right). \end{array}$$
where $G_{prod}$ (mg/kg/min) and $G_{up}$ (mg/kg/min), represents the production and uptake rates of hepatic glucose produced by exercise and $I_{e}$ (μU/mL/min) to the rate of insulin clearance from the circulatory system because of exercise-induced physiological changes.

Percentage of the maximum oxygen consumption rate:(8)$$\frac{dPVO_{2}^{max}\left(t\right)}{dt}=-0.8PVO_{2}^{max}\left(t\right)+0.8u\left(t\right).$$

The percentage of the maximum oxygen consumption rate ($PVO_{2}^{max}$) is a factor to indirectly measure the exercise levels experienced by the individual. Therefore, Roy and Parker [30] takes into account that a person in a basal state consumes about 8% of $PVO_{2}^{max}$.

Critical threshold in energy expenditure:(9)$$A_{TH}=-1.152{\left(u\left(t\right)\right)}^{2}+87.471u\left(t\right).$$

The critical threshold in energy expenditure ($A_{TH}$) is related to the function of intensity and duration of exercise. When the energy expenditure is greater than the value of $A_{TH}$, the glycogenolysis rate ($G_{gly}$) begins to decrease.

Integrated exercise intensity:(10)$$\frac{dA(t)}{dt}=\left\{\begin{array}{cc} u(t) & u(t)>0\\ -\frac{A(t)}{0.001} & u(t)=0 \end{array}\right.$$

The integrated exercise intensity (*A*), is calculated according to the percentage of the exercise intensity above the basal level (*u*). When (u(t)>0) it is considered that there is the presence of physical activity and when (u(t)=0) the beginning of recovery period is considered, which marks the replenishment beginning of glycogen stores through a continuous elevation in the hepatic gluconeogenesis rate.

Glycogenolysis dynamics during prolonged exercise:(11)$$\frac{dG_{gly}\left(t\right)}{dt}=\left\{\begin{array}{cc} 0 & A\left(t\right)<A_{TH}\\ K & A\left(t\right)\geq A_{TH}\\ -\frac{G_{gly}\left(t\right)}{T_{1}} & u(t)=0 \end{array}\right.$$

$G_{gly}$ (mg/kg/min), is the decrease in the rate of glycogenolysis during prolonged exercise because depletion of liver glycogen stores. As long as the *A* value remains below the critical threshold value ($A_{TH}$), there is still enough glycogen available to maintain a sufficient hepatic glucose release rate. On the contrary, when *A* reaches $A_{TH}$, the rate of change of the glycogenolysis rate decreases at a rate given by *K*, as a consequence of the depletion of liver glycogen reserves. Finally, when (u(t)=0), the recovery period of liver glycogen stores begins.

#### 2.1.3. System Inputs and Disturbances

Alimentary Bolus 

In order to contemplate the effect of those carbohydrate (CH) intakes of slow ($D_{GS}$) and fast ($D_{GF}$) absorption (g), the Equation (Equation 12) was proposed. Which is based on what was proposed by Rodríguez-Herrero et al. [31], considering separately each of two types carbohydrates effects and integrating them below.
(12)$$d\left(t\right)=D_{GS}\frac{r_{1}r_{2}}{r_{2}-r_{1}}\left(e^{-r_{1}t}-e^{-r_{2}t}\right)+D_{GF}\frac{r_{3}r_{4}}{r_{4}-r_{3}}\left(e^{-r_{3}t}-e^{-r_{4}t}\right).$$

Percentage of Exercise Intensity above Baseline 

Based on the works of Caballero et al. [32], Gilgen et al. [33] and Kang et al. [34], where the physical activity intensity is related to the percentage of subject’s maximum heart rate. In Equation (Equation 13) a mathematical relationship was proposed to calculate the exercise intensity percentage above the basal level (*u*), with respect to heart rate ($H_{R}$, bpm) and maximum heart rate ($H_{Rmax}$, bpm) of subject
(13)$$u\left(t\right)=\frac{100H_{R}\left(t\right)}{H_{Rmax}}.$$

It should be emphasized that this model is limited to people with normal weight. Because the proposed model does not consider the effect of the degradation of glucose homeostasis derived from an overweight person. Since weight is used as a factor in the rate of glucose reduction derived from physical activity.

According to the mathematical representation, in Table 1, the rest description of parameters contemplated in this mathematical model are presented.

To give another view of the proposed mathematical model, Figure 1 presents the diagram of the mathematical representation of glucose homeostasis in healthy people. Where the representation defined by Orozco-López et al. [15] is described in red, according to the dynamics of the glucose homeostasis in healthy people (*I*, *X* and *G*). In green the effect dynamics on the glucose homeostasis due to exercise ($G_{prod}$, $G_{up}$, $I_{e}$ and $PVO_{2}^{max}$) proposed by Roy and Parker [30]. Finally, in blue, the mathematical representations of the alimentary bolus (*d*) and exercise intensity percentage (*u*) proposed are illustrated.

### 2.2. Experimentation Definition

For the purpose of generating conductive data on the variation in the dynamics of glucose homeostasis for persons with sedentary tendencies and physically active persons in relation to the effects of diet and physical activity. In this study, data were collected on personal factors, dietary habits and daily routine of physical activity performed by a group of 20 Mexican individuals (9 women, 11 men) with different occupations between 22 and 75 years of age. In order to expand the study sample, the database of the National Health and Nutrition Survey (ENSANUT, 2018) was used, from which the data of 69 Mexican individuals (36 women, 33 men) between the ages of 20 to 60 years (age, weight, diet, frequency and level of physical activity) were obtained. This resulted in a sample with real experimental data of 89 people, and also generated a sample of 200 virtual patients based on the Monte Carlo approach, (50% of the sample are people with sedentary habits and the rest with regular physical activity) to evaluate the results in the performance of the proposed mathematical model.

The Monte Carlo approach was used as a method to generate a range of random scenarios from a bounded initial data set by making variants focused on model validation in simulation (For more information see Landau and Binder [35]).

#### 2.2.1. Personal Factor Measurements

In Table 2, the characteristics of the populations studied are shown, in column 3 own sample and in column 4 those obtained through the ENSANUT 2018. Age, height, weight, body mass index, fat percentage, muscle percentage, visceral fat percentage, resting heart rate, and plasma glucose concentration were the factors taken into account. In the particular case of the sample of 20 people, the glucose concentration of some participants was higher than 85 mg/dL. Due to limitations in the ENSANUT 2018 database, it was not possible to have all the parameters obtained through the sociodemographic questionnaires conducted on the sample of 20 people. It should be noted that the data obtained from ENSANUT were sufficient to feed the proposed model and to make a contrast between the population of 20 people. The reference factors were height, percentages of fat, muscle and visceral fat, and resting heart rate.

#### 2.2.2. Instrumentation

To obtain an approximation in the basal values of blood glucose concentration, a measurement of fasting glucose concentration was carried out on each person, for this an Accu-Chek^®^ Active blood glucose monitor was used, a Puncture system with Accu-Chek^®^ Softclix lancets, in addition to the Accu-Chek^®^ Active test strips. The measurements of weight, body mass index, fat percentage, muscle percentage and visceral fat percentage, an Omron^®^ HBF-514C digital scale was used. A CITIZEN^®^ CH-308B blood pressure monitor was used for heart rate measurements at rest, finally a flexometer was used to measure the height of each participant.

#### 2.2.3. Sectioning

In order to study the evolution of the dynamics of blood glucose concentration in people who present regular physical activity and those who present sedentary behavior throughout the day, the sample of participants analyzed was divided into two sectors: first group of physically active people and the second of people with sedentary habits. In accordance with what was previously described Ceron et al. [36], participants were considered to have sedentary physical activity when the values of the metabolic equivalent per task (MET) were equal or less than 1.5 METs. For each of the participants analyzed, the time that physical activity greater than 1.5 METs was performed throughout the activity period in a day was calculated, if the time was greater than or equal to a 8 h it is considered that the participant had a physically active life and consequently when the value was less than 8 h it was considered that the participant had sedentary tendencies. An example of the analysis for this result is presented in the Table 3, where each of the participants in the 20 people sample is classified according to their occupation and the hours dedicated to physical activities greater than 1.5 METs.

#### 2.2.4. Metabolic Equivalent of Task

The approximate value of the metabolic equivalent per task (MET) developed in each subject was calculated based on the proposed mathematical representation (Equation 14). This representation is a Gaussian function that was parameterized based on the values of the relationship between the heart rate percentage and the number of METs developed per activity performed presented in Caballero et al. [32].
(14)$$MET=a*e^{-{\left(\frac{u(t)-b}{c}\right)}^{2}}$$

Figure 2, presents the results of the estimation based on the data presented by Caballero et al. [32], where the mean square error value was 0.1669. Parameters a, b and c are considered as adjustment parameters, a = 10.25 MET, b and c are parameters related to the system input (% $H_{R}$ developed), b = 89.26% $H_{R}$ and c = 62.36% $H_{R}$.

#### 2.2.5. Generation of a Population Sample based on the Monte Carlo Approach

In order to increase the number of the population sample, 200 virtual patients were generated divided into 100 physically active people (50% women and 50% men) and 100 sedentary people (50% women and 50% men). The following routine and physiological data were generated for each of the patients: height, age, weight, body mass index (BMI), basal glucose level, amount of food intakes, number and time of intakes, intensity and time of physical activity.

In the case of the personal factors of the virtual patients generated, Table 4 describes the general average and its standard deviation for each sector of the population analyzed, sectioning by the gender of each patient.

The generation of virtual patients was developed based on the scheme in Figure 3, where it begins by defining the gender and type of occupation of the patient, then based on gender and occupation, the random data of age and height are generated. After that, a body weight value is generated, with the height and weight the body mass index (BMI) is calculated, from the BMI random values of basal glucose, the intensity of physical activity and the amount of carbohydrates consumed are obtained throughout the day. The number of meals and time of ingestion is defined based on the amount of carbohydrates consumed.

In this study, 5 physical activities per virtual patient were contemplated, these activities were developed after a random time no longer than 3 h after each intake. Based on Caballero et al. [32], the intensities of each physical activity developed by each of the sectors analyzed were contemplated, in the case of physically active people, walks between 55 to 100 m/min were contemplated and as an activity of greater effort physical runs of 130 m/min. For sedentary people, office activities were considered, such as the use of a computer and working with documentation while the person is sitting, as an activity of greater physical effort, domestic activities such as sweeping were contemplated.

## 3. Results

### 3.1. Effect of Primary and Secondary Factors on the Behavior of Glucose Homeostasis

Glucose concentration is closely related to diet and activity, for this reason food intake has been considered as the main factor in the variations of glucose concentration in the organism, since this factor produces a high degree of change in glucose dynamics. On the other hand, physical activity as well as stress and fatigue have been considered as secondary factors, since in comparison to diet, the impact on the dynamics of glucose concentration turns out to be much smaller. Figure 4 shows the behavior of glucose concentration in a healthy person with regular physical activity according to diet and physical activity factors. From this figure, the correlation between primary and secondary factors in the variation of blood glucose concentration in healthy people is demonstrated. Diet being a factor that increases blood glucose levels quickly and highly, while the impact of physical activity causes a slow and minimal decrease in the behavior of blood glucose.

### 3.2. Dynamic Behavior of Glucose Homeostasis

As an example case in Figure 5 and Figure 6, the isolated case shows the dynamics of glucose homeostasis in participant 14, who in this case was an office worker under the work from home scheme. For her, four food intakes were contemplated (breakfast, lunch, snack and dinner), first intake had a carbohydrate amount of 47.6 g (only slow absorbing carbohydrates), composed of a vegetables portion, two slices of bread, lean meat, 1/3 of an avocado and 1 water glass, developing at 10:00 am; subsequently, second intake occurred at 2:00 p.m. with an amount of 41 g (only slow absorbing carbohydrates) consisting of a vegetables portion, 2 corn *tortillas*, lean meat, 1/3 of avocado and 2 pure water glasses; between 6:00 p.m., 16.3 g from an apple and a small handful of almonds were ingested; finally at 9:00 p.m., last intake was developed with an amount of 39.7 g (16.8 g of slow absorption and 22.9 g of rapid absorption), consisting of 1 slice of bread and a liquid yogurt.

Regarding physical activity, the following was contemplated: work on the computer from 8:30 to 9:30 a.m., then from 9:30 to 10:00 a.m., domestic activities were performed. After the first meal, a walk was taken until 11:30 a.m. At the end of the activity, various activities were carried out which can range from computer work to domestic activities ending until the second lunchtime. Later, at 2:30 p.m., activities such as reading or checking social networks were carried out. From 4:00 to 4:30 p.m a short nap was performed, at the end of the nap the participant worked on the computer until 6:00 p.m., during this time a snack was ingested between taking a walk, which ended until 8:00 p.m. where domestic activities were performed until the last intake performed at 9:00 p.m., finally from 9:30 p.m. until midnight activities ranging from watching TV to the use of the smartphone were performed.

Figure 5, presents the graphical results in the variation of glucose in participant 14 throughout the day. This figure is divided into three sections, the upper section presents the variation of glucose considering the physical activity performed throughout the day (blue color) and in contrast presents the variation of glucose without considering the effects of physical activity (red dashed line); in the middle and lower part of the figure are presented the dynamics of food intake (blue color) and physical activity performed based on oxygen consumption in the system (orange color), the effect of both factors is reflected in the glucose concentration, which rises to values above 90 mg/dL after each intake and can reach values close to hypoglycemia (70 mg/dL, as defined in Bequette et al. [37]) due to the over exposure of physical activity as can be seen in the 20th hour of simulation.

In Figure 6, physical activity effect on the glucose homeostasis is illustrated. The upper part shows the glucose consumption rate over a 24 h period in the face of variations in the intensity of physical exercise. In the middle part the effect produced in hepatic glucose production is shown, finally in the lower part the insulin removal rate due to physical activity is shown. As can be seen, there is a marked relationship between physical activity and the variation of both insulin and glucose in the short term, which depending on the intensity, the required demand will be greater.

### 3.3. People with Regular PHYSICAL activity

From the sample of 20 healthy persons, a total of 7 persons were considered physically active (2 of them women) and in relation to the ENSANUT subsample, a total of 39 physically active persons were obtained (24 of them women). The sectioning of this population was based in whether the person spent more than 8 h of the day developing activities above 1.5 METs. Figure 7 presents a comparison between the glucose dynamics in the population of physically active people from the sample of 20 healthy people (item a, left side of the figure) and the ENSANUT 2018 subsample (item b, right side of the figure). In both cases there turns out to be some degree of discrepancy between each of the individuals analyzed since in each of the cases the times of food intake and physical activity vary from person to person, but generally remain at values between 80 and 100 mg/dL except for some cases where there were amounts of carbohydrates consumed above the normal values for the population and therefore glucose levels turned out to be higher than normal. Or in the opposite case where the amount of intake was lower than average leading to glucose concentration values below 80 mg/dL. The behavior of physical activity was very similar in most cases in both samples, with mean values of 20% of oxygen consumed in the sample of 20 healthy people and 50% in the ENSANUT subsample.

Regarding the data generated with the Monte Carlo approach, the behavior of glucose homeostasis was analyzed in 100 physically active virtual patients (50 of them women) over a 24-h period, simulating by means of 5 physical activities and 3 to 4 random food intakes in a normal day in the daily life of these people. Figure 8, represents the behavior of blood glucose for the case of 50 physically active men before the effect of the factors of diet and physical intensity performed. The behavior of glucose turns out to oscillate between values of 80 and 120 mg/dl throughout the day, the highest values are the product of each of the intakes made by the virtual patient and the decrease in them is the product of the effect of insulin and physical activity developed throughout the day. Food intake tends to be varied according to the time it develops, but with similar values between them. Because they are physically active people, the maximum percentage of oxygen consumed tends to oscillate between 8 and 80% of its totality, generally remaining at values of 30% of its totality throughout the period of activity of the person.

### 3.4. People with Sedentary Habits

The rest of both populations analyzed were classified as having a sedentary tendency because they spend less than 8 h of the day performing activities above 1.5 METs. The number of people in the population of 20 healthy people was 13 participants (6 of them men), while that of the ENSANUT subsample was 30 participants (18 men). Figure 9 shows the comparison between samples in the variations of glucose concentration throughout the day in relation to the effects of carbohydrate intake and the physical activity performed by people with sedentary tendencies. Compared to the physically active population, the behavior for glucose in the population of 20 people is more volatile due to the variability in the amount of carbohydrates consumed, which tend to be higher than in the case of the active population. While in the ENSANUT subsample, it turns out to be more constant due to the fact that two intakes are generally contemplated throughout the day. The physical activity, on the other hand, in both cases turns out to oscillate between values of 10 to 30% of the oxygen consumption, presenting peculiarities in some cases where due to the activity performed the oxygen consumption oscillates between values of 45 to 80%, being generally the only activity higher than 1.5 METs performed by these people throughout the day.

Based on the data generated from the Monte Carlo approach, the behavior of glucose homeostasis was analyzed in 100 virtual patients with sedentary tendencies (50 of them men) over a period of 24 h, simulating by means of 5 physical activities of the sedentary type (use of mobile devices, sitting jobs and housework) quantified based on what was proposed in Caballero et al. [32], in addition to 3 to 4 random food intakes on a normal day in the daily lives of these people. Figure 10 shows the case of women with sedentary habits based on the Monte Carlo approach, turning out to be similar to that proposed in the case of physically active virtual patients, but in this case having the peculiarity that the maximum percentage of oxygen consumed turns out to oscillate between 8 and 30% of the total maximum consumption, sedentary activities prevailing at all times throughout the day.

### 3.5. Comparison between the Variation of Glucose Homeostasis of the Analyzed Profiles

The development of physical activity to regulate blood glucose levels is an important factor to take into account, because in the case of people who perform physical activities on a regular basis it is possible to avoid peaks in glucose levels derived from dietary intakes in addition to leading promptly to stable glucose levels, greatly improving the performance of glucose homeostasis.

Figure 11, presents a graphic example of the previously mentioned, illustrating the effect of dietary intake and physical activity in a person with sedentary tendencies (blue line) and a physically active one (red line). The figure presents two types of contrast: the first one presents the variation of glucose when physical activity is considered in the mathematical equations (continuous lines) and when it is omitted (dashed lines) in both analyzed profiles; on the other hand, the second contrast presents the difference between the dynamics of glucose in a person with sedentary tendencies and one who is physically active on a regular basis. In the case of the person who has sedentary tendencies, the effect of physical activity on the decrease in glucose is discrete, giving a similar behavior between the glucose dynamics when physical activity is considered and when it is not considered. The opposite is happening in the physically active person, where the decrease in glucose levels derived from physical activity is considerable, thus avoiding a precipitous increase in glucose derived from a dietary intake.

## 4. Discussion

In this work a mathematical model focused on developing numerical approximations of the dynamics of glucose homeostasis in healthy individuals under different in-silico experimental scenarios was presented. Based on the proposed results, the following points were generated for discussion:

Given the close relationship between mathematical models and in-silico experimentation, these are a complementary tool in the development of numerical approximations on scenarios of diverse case studies in physiological phenomena. In this particular case, by means of mathematical representations it was possible to generate virtual populations on the variation of glucose homeostasis in different population sectors. Thus obtaining correlations between the increase and decrease of glucose levels derived from the effect of factors such as diet and physical activity.

Due to the limitations of the experimentation to give validity to the proposed studies, in this work the Monte Carlo methodology was used, which is a strategy to generate a large number of probabilistic scenarios, in which parametric variations bounded to realistic limits of the analyzed environment are contemplated to generate virtual populations, similar to the process carried out in the works of Lewis and Mueller [38] and Bekisz et al. [39]. According to this methodology, it was possible to take into account in the proposed virtual populations the physiological variation of each of the patients, the frequencies and time in the development of intakes and development physical activity in addition to the amount from carbohydrates consumed and physical intensity developed. Thus generating realistic scenarios applicable to population studies where the amount of conductive data is tiny and an expansion of the sample is necessary.

Regarding that the main features of the proposed model are focused on modeling the effects of exogenous factors such as diet and intensity of physical activity on the variation of glucose levels. In this work based on three types of population samples (a) Sample collected from 20 healthy people (b) Subsample of 69 people collected from ENSANUT 2018 and (c) Sample of 200 virtual patients based on Monte Carlo a comparative study was developed between the glucose variation in front of behaviors activities in sedentary and physically active people. According to the results, the frequency and quantity of food consumption is the main factor in the elevation of glucose, which in conjunction with regular physical activity is achieved to have adequate glucose levels for the proper functioning of vital functions of the body. On the other hand, a poor quantity and low frequency of food destined in the diet together with a prolonged amount of physical activity promote low glucose conditions in the person.

However, it is true that these results in health professionals turn out to be quite intuitive without the need to develop this type of methodology. The main contribution of this work is to demonstrate the ability of virtual populations to generate in-silico approximations similar to those obtained through in-vivo studies. Generating conductive data that are only achievable through in-vivo experiments such as the time in which glucose levels remain elevated by diet or those where the person is more vulnerable due to an insufficiency in food and a high degree of physical activity developed. Being able to obtain not only this type of information but also more specific information according to the needs of the study developed for the generation of prevention alternatives in the development of type 2 diabetes. Our working group intends to delve into these types of models in order to generate tools that can be used directly by health professionals.

Contrasting the proposed work in relation to the most current works in the literature, it was observed that the main particularity of this work is focused on the study of the effects of factors such as diet and physical activity in healthy people, being the areas of greatest interest those focused on diabetes. Examples of some of the work consulted are Ahmad et al. [40], developing virtual patient populations capable of simulating blood glucose metrics in patients with T1DM; Krishnamoorthy et al. [41] applying in-silico experimentation on 50 virtual patients with T2DM to find optimal insulin dose values; Smaoui et al. [42] developing a platform for the development and in-silico experimentation of experimental protocols in patient with T1DM.

The ability of the mathematical model obtained to generate a dynamic approximation of the behavior in glucose homeostasis is remarkable despite the minimal number of parameters used as in the case of the ENSANUT database. The determination of the mathematical model is remarkable since it has an impact on the analysis of in-silico population studies that allows validating its use for in vivo activities without exposing patients in the research phase.

## 5. Conclusions

A mathematical representation of glucose homeostasis in relation to dietary and physical activity factors in healthy individuals is presented in this work. The particularity of this representation focuses on the incorporation of an alternative mathematical representation of dietary intake that considers the effect of slow and fast absorbing carbohydrates and a mathematical representation of the intensity of physical activity performed based on the metabolic equivalent per task. To demonstrate the potential of this mathematical representation, a comparison between the dynamics of glucose homeostasis in people with sedentary tendencies and those who are physically active was proposed. The present work aims to be a modeling tool capable of simulating in a simple and realistic way the behavior of glucose homeostasis in healthy people under daily life conditions when considering the effects derived from physical activity and diet. This tool is proposed as an alternative to generate data conducive to the development of biomedical and epidemiological studies on glucose homeostasis.

## Figures and Tables

**Figure 1 ijerph-19-00716-f001:**
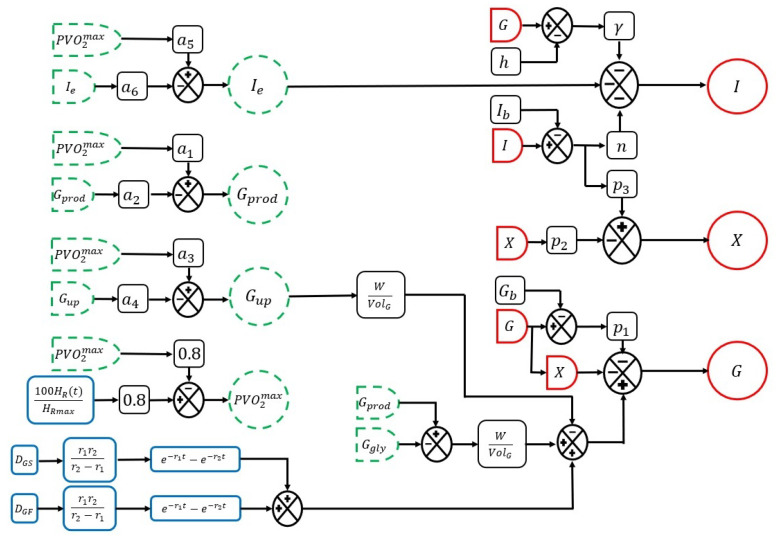
Proposed mathematical representation of glucose homeostasis in healthy people. The time-varying equations are enclosed by a circle and labels are placed on them to make the diagram more understandable. The expressions enclosed in a black box are the system parameters and those enclosed in a blue box represent the inputs and disturbances of the system.

**Figure 2 ijerph-19-00716-f002:**
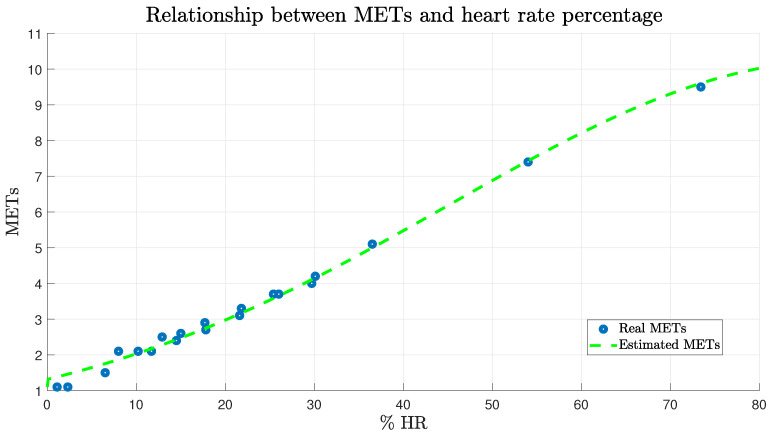
METs estimation based on the heart rate percentage. Graphical result between the estimation of the relationship between METs and heart rate percentage in relation to the data of Caballero et al. [32].

**Figure 3 ijerph-19-00716-f003:**
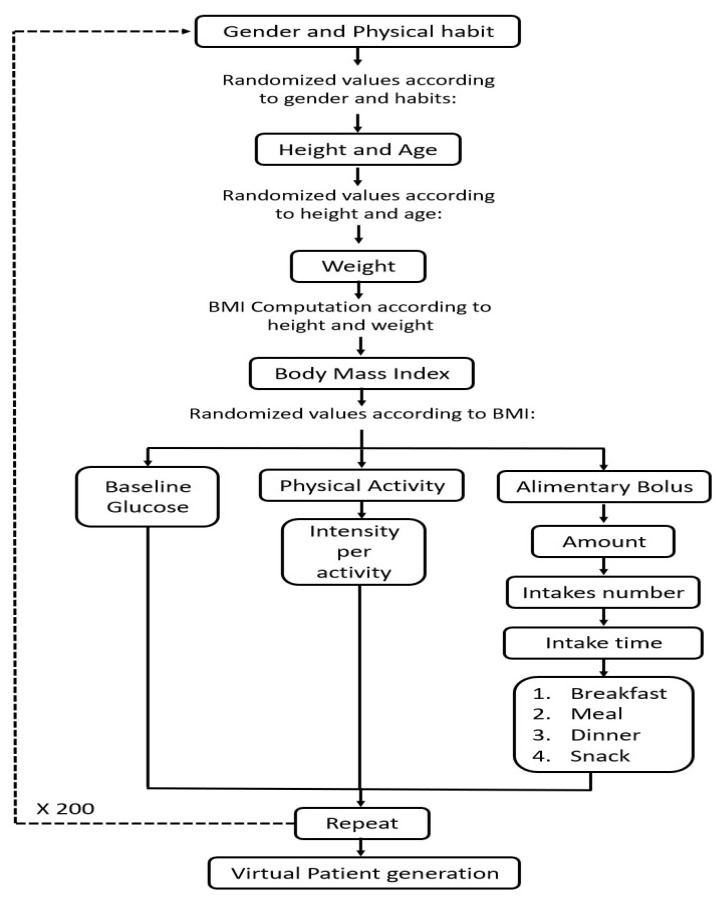
Generation of the virtual population. Methodology developed for the generation of 200 virtual patients with 100 of them having sedentary habits and the rest having regular physical activity.

**Figure 4 ijerph-19-00716-f004:**
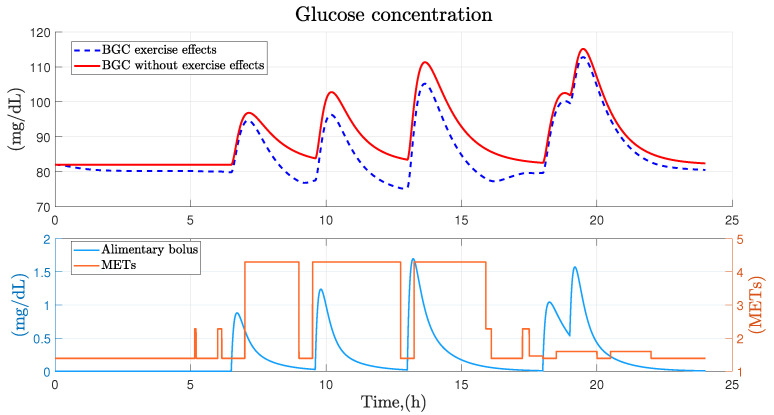
Effect of diet and physical activity on glucose concentration. Graphical visualization of the effect from diet and physical activity on the dynamics of blood glucose concentration.

**Figure 5 ijerph-19-00716-f005:**
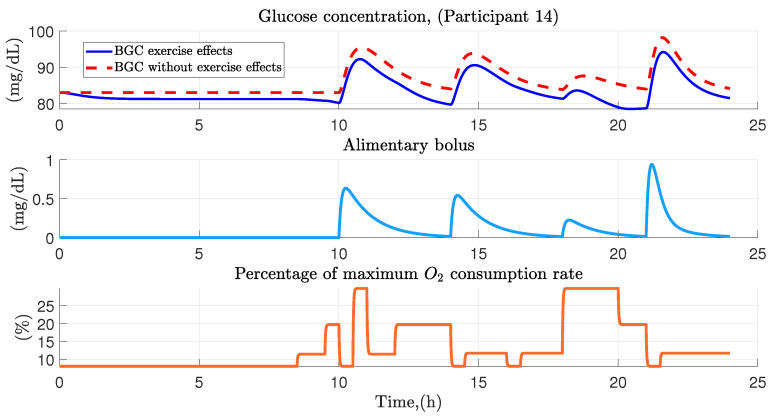
Variation of blood glucose concentration in the participant 14. Graphical representation of the effect of physical activity and diet in participant 14, placing in a dashed line the glucose dynamics without considering physical activity.

**Figure 6 ijerph-19-00716-f006:**
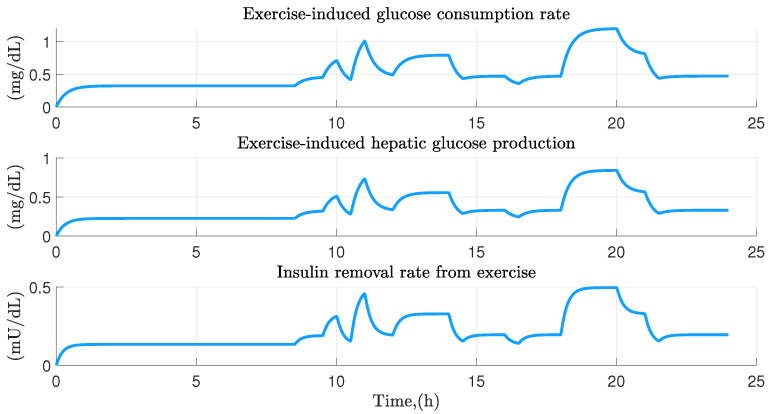
Exercise effect on the glucose homeostasis of the participant 14. Graphical representation of the reduction in glucose and insulin levels derived from the physical activity performed.

**Figure 7 ijerph-19-00716-f007:**
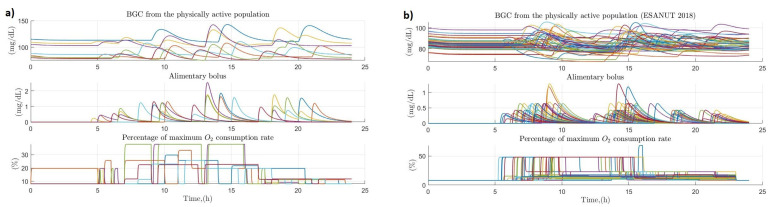
Effect of primary and secondary factors on blood glucose in physically active populations. (**a**) Effect of diet and physical activity on blood glucose concentration in physically active persons (sample of 20 healthy persons); (**b**) Effect of diet and physical activity on blood glucose concentration in physically active people (ENSANUT 2018 subsample).

**Figure 8 ijerph-19-00716-f008:**
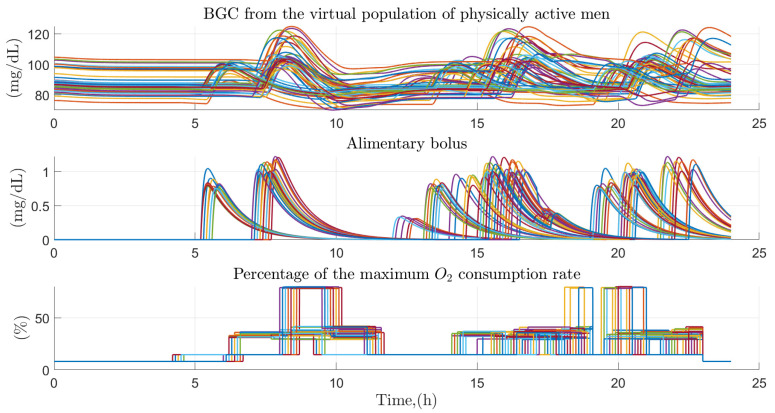
Effect of primary and secondary factors on the virtual population of physically active men. Effect of diet and physical activity on glucose concentration in the virtual population of 50 physically active men.

**Figure 9 ijerph-19-00716-f009:**
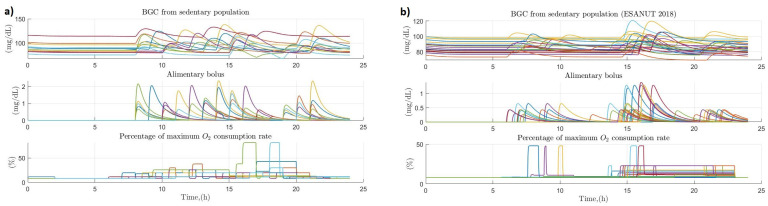
Effect of primary and secondary factors on blood glucose in sedentary populations. (**a**) Effect of diet and physical activity on glucose concentration in a sample of people with sedentary habits (sample of 20 healthy people); (**b**) Effect of diet and physical activity on glucose concentration in a sample of people with sedentary habits (ENSANUT 2018 subsample).

**Figure 10 ijerph-19-00716-f010:**
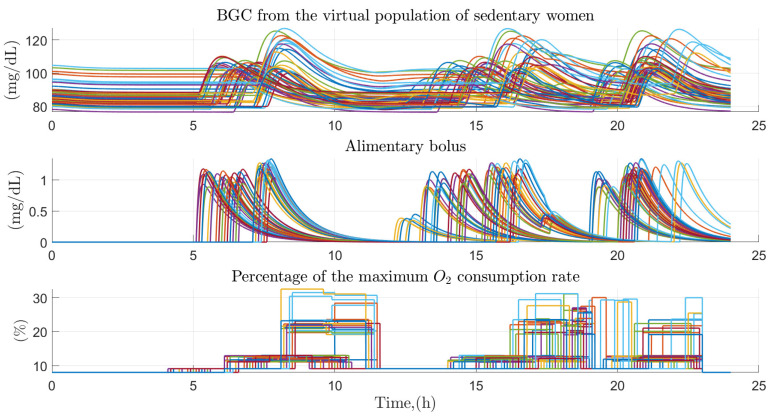
Effect of primary and secondary factors on the virtual population of sedentary women. Effect of diet and physical activity on glucose concentration in a virtual population of 50 women with sedentary habits.

**Figure 11 ijerph-19-00716-f011:**
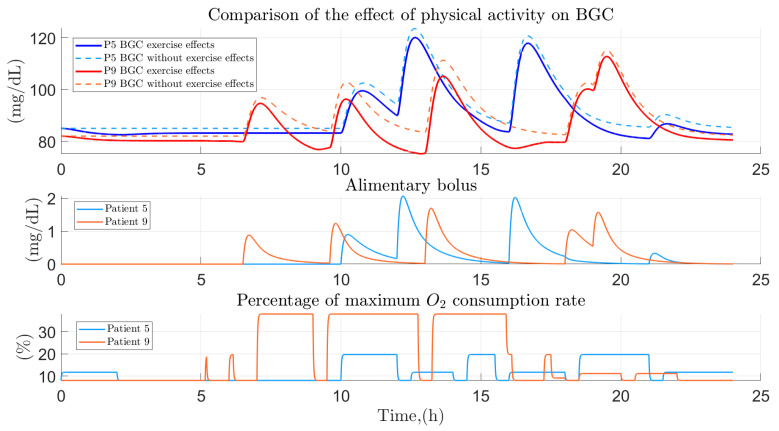
Comparison between the glucose dynamics of a physically active person and one with sedentary tendencies. Contrast between the effect of exercise on glucose lowering between a physically active person and a person tending to be sedentary.

**Table 1 ijerph-19-00716-t001:** Parameters description of the Extended Bergman minimal model.

Symbol	Units	Description
$p_{1}$	(1/min)	Insulin independent rate constant
$p_{2}$	(1/min)	Rate for decrease in tissues glucose uptake ability
$p_{3}$	[(μU/mL)/min^2^]	Insulin-dependent increase
η	(1/min)	First-order decay rate for insulin in blood
*h*	(mg/dL)	Glucose threshold above β cells release insulin
γ	[(μU/mL)/(min^2^ (mg/dL))]	Insulin release rate from pancreatic β cells
*W*	(kg)	Weight of the subject
$Vol_{G}$	(dL)	Glucose distribution volume
$a_{1}$	(mg/kg min^2^)	$G_{prod}$ dynamic constant
$a_{2}$	(1/min)	$G_{prod}$ dynamic constant
$a_{3}$	(mg/kg min^2^)	$G_{upt}$ dynamic constant
$a_{4}$	(1/min)	$G_{upt}$ dynamic constant
$a_{5}$	(μU/mL min)	$I_{e}$ dynamic constant
$a_{6}$	(1/min)	$I_{e}$ dynamic constant
$T_{1}$	(min)	Glycogenesis time constant
$r_{1}$	(1/min)	CH slow absorption parameter
$r_{2}$	(1/min)	CH slow absorption parameter
$r_{3}$	(1/min)	CH fast absorption parameter
$r_{4}$	(1/min)	CH fast absorption parameter

**Table 2 ijerph-19-00716-t002:** Personal characteristics of the participants involved in the experimentation.

		Own Sample	ENSANUT, 2018
Demographic Variable	Unit	Mean ± SD	Mean ± SD
Age	Years	42.75 ± 16.35	36.10 ± 11.35
Height	m	1.68 ± 0.09	-
Weight	kg	70.00 ± 13.24	71.86 ± 10.97
Body mass index	Kg/m^2^	24.58 ± 4.05	-
Fat percentage	-	27.88 ± 9.04	-
Muscle percentage	-	32.10 ± 6.74	-
Visceral fat percentage	-	6.6 ± 2.85	-
Heart rate (rest)	bpm	75.4 ± 7.32	-
Glucose concentration	mg/dL	91.55 ± 11.78	85.37 ± 5.06

**Table 3 ijerph-19-00716-t003:** Occupation of participants and activity time.

# Participant	Occupation	Time (h)	# Participant	Occupation	Time (h)
1	Retired	5.10	11	Student	0.76
2	House keeper	9.69	12	Professional	6.63
3	Office worker	2.80	13	House keeper	7.65
4	Student	0.51	14	Office worker	6.12
5	House keeper	5.61	15	Retired	8.16
6	House keeper	14.78	16	Retired	7.14
7	Factory worker	9.93	17	Office worker	3.57
8	Factory worker	9.42	18	Professional	5.81
9	Factory worker	8.72	19	Office worker	5.61
10	Student	1.53	20	Professional	8.16

**Table 4 ijerph-19-00716-t004:** Personal factors of the developed virtual population.

		Mean ± SD			
Factor	Unit	Physically Active		Sedentary	
		Men	Women	Men	Women
Age	years	42 ± 14	41 ± 12	43 ± 14	40 ± 12
Basal glucose	mg/dL	88.06 ± 5.25	88.63 ± 7.35	89.90 ± 7.81	90.02 ± 7.05
BMI	kg/m^2^	23.59 ± 2.4	23.79 ± 3.12	25.67 ± 3.4	25.03 ± 3.22
CHOs quantity	g	300 ± 27	279 ± 34	305 ± 35	288 ± 32
Height	m	1.69 ± 0.06	1.59 ± 0.07	1.69 ± 0.06	1.60 ± 0.06
Weight	Kg	67.39 ± 6	63.02 ± 7	73.02 ± 9	64.68 ± 8

## Data Availability

The data corresponding to the ENSANUT 2018 database can be found at the following link: https://ensanut.insp.mx/encuestas/ensanut2018/descargas.php. The data of the 20 healthy participants can be requested from the corresponding author (d18ce081@cenidet.tecnm.mx) because they are not yet public.

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
