# Peer review of "Dynamic of Glucose Homeostasis in Virtual Patients: A Comparison between Different Behaviors"

_ijerph, 2022, doi:10.3390/ijerph19020716_

Round 1
Reviewer 1 Report
Comments
The manuscript presents a mathematical representation of the dynamics of glucose homeostasis when affected by food and physical activity levels. The aim is to help in prevention strategies aiming to reduce the risk of type 2 diabetes in this population. My main concern here is the novelty. There have been similar studies in the literature that included healthy people, what is your study adding? This requires a strong argument to demonstrate the relevance of this research.
It doesn’t look like you differentiated between normal weight and overweight “actual” participants. How many overweight and normal weight individuals have you recruited? In addition to overweight people being at higher risk of type 2 diabetes compared to their normal weight counterparts, please note that we are deviating from referring to overweight people as “healthy”. Only those with normal BMI are considered healthy.
Remember many of the readers who have research interests in nutrition and diabetes have no sound understanding of mathematical models. I suggest you explain in a simplified way how health professionals can use this information in research/practice.
Further comments:
Lines 26-18: “It is important to seek alternatives that improve the management and prevention of the disease in the country, through projections that provide more information on sectors vulnerable to suffering from this condition”. Considering the country has done significant investments in diabetes, why the alternative? Please rephrase.
Line 161: Please specify whether all participants had normal glucose levels. From your standard deviation, some people might have had high glucose levels.
Author Response
Response to Reviewer 1 Comments
Point 1: The manuscript presents a mathematical representation of the dynamics of glucose homeostasis when affected by food and physical activity levels. The aim is to help in prevention strategies aiming to reduce the risk of type 2 diabetes in this population. My main concern here is the novelty. There have been similar studies in the literature that included healthy people, what is your study adding? This requires a strong argument to demonstrate the relevance of this research.
Response 1: In order to present a solid argument to your question, the answer has been divided into the following points:
Similar studies in the literature:
Indeed, in the literature there are studies that relate these factors to glucose variation, but the main characteristic of these studies is that they are based on experimental developments where the experimental time and the number of samples are generally limited. Now, from the mathematical point of view, there is a large amount of work focused on describing the behavior of glucose homeostasis, but generally the work is focused on studying the case of insulin-dependent people due to the insulin control applications that can be developed. In the case of glucose homeostasis in healthy individuals, information from a mathematical approach is extremely scarce.
To strengthen this argument, in the introduction (L45 - L62) and discussion (L413 - L421) section, sentences were added in which reference is made to the works focused on glucose homeostasis from a mathematical approach developed in the last two years. Thus, the main focus of these works is directed to studies in insulin-dependent individuals.
Novelty of work:
The novelty of this work is focused on presenting a mathematical tool with the ability to model the effect of exogenous factors (in this particular case, diet and physical activity) on the variation of glucose levels in healthy people over time, and thus generate numerical data that allow us to know the effect of glucose variations in a prepathogenic state.
Relevance of the research:
The relevance of this work is focused on the wide range of applications that can be carried out with this mathematical model, such as:
- From the point of view of glucose control in insulin-dependent patients, the proposed model can be used as a complement in the development of control strategies in artificial pancreas systems. Serving as a reference signal in glucose regulation.
- Development of projections of the dynamics of glucose homeostasis in a population of healthy people. Generating indications on those sectors vulnerable to present anomalies in glucose control derived from an adequate management of food and physical activity developed.
- Consideration of the high economic costs and difficulties involved in the development of continuous glucose monitoring in population studies to know the state of glucose homeostasis. The proposed mathematical tool generates numerical approximations of the results obtained from this type of studies, thus reducing costs and time spent in the study.
Point 2: It doesn’t look like you differentiated between normal weight and overweight “actual” participants. How many overweight and normal weight individuals have you recruited? In addition to overweight people being at higher risk of type 2 diabetes compared to their normal weight counterparts, please note that we are deviating from referring to overweight people as “healthy”. Only those with normal BMI are considered healthy.
Response 2: Thanks to the reviewer for the comment. In response, we comment that it is indeed interesting to consider weight and obesity, however in the contribution of the work presented here is not considered as a direct variable in the model, but we believe that other work will be interesting to make a mathematical model that considers the impact of this variable directly. However, we inform you that in the analysis we omitted those people with obesity in whom there is a degradation of glucose regulation. Also, weight is used as a factor in the rate of glucose reduction derived from physical activity. Therefore, the model is limited to people without obesity.
Point 3: Remember many of the readers who have research interests in nutrition and diabetes have no sound understanding of mathematical models. I suggest you explain in a simplified way how health professionals can use this information in research/practice.
Response 3: Based on your comment, the authors have added the following paragraph in the discussion section that emphasizes your comment.
“(L402 - L412) However, it is true that these results in health professionals turn out to be quite intuitive without the need to develop this type of methodology. The main contribution of this work is to demonstrate the ability of virtual populations to generate in-silico approximations similar to those obtained through in-vivo studies. Generating conductive data that are only achievable through in-vivo experiments such as the time in which glucose levels remain elevated by diet or those where the person is more vulnerable due to an insufficiency in food and a high degree of physical activity developed. Being able to obtain not only this type of information but also more specific information according to the needs of the study developed for the generation of prevention alternatives in the development of type 2 diabetes. Our working group intends to delve into these types of models in order to generate tools that can be used directly by health professionals.”
Further comments:
Point 4: Lines 26-18: “It is important to seek alternatives that improve the management and prevention of the disease in the country, through projections that provide more information on sectors vulnerable to suffering from this condition”. Considering the country has done significant investments in diabetes, why the alternative? Please rephrase.
Response 4: The authors thank you for your comment and in response to it, the authors in (L19 - L23) added a previous sentence specifying the suggested aspects to be included in the sentence.
“In Mexico, the National Health and Nutrition Survey reported that 8,542,718 people over 20 years of age had a previous medical diagnosis of type 2 diabetes mellitus (T2DM) [1]. For this reason, it is necessary to have a better understanding of the effects that factors such as diet and physical activity have on glucose homeostasis.”
Point 5: Line 161: Please specify whether all participants had normal glucose levels. From your standard deviation, some people might have had high glucose levels.
Response 5: Regarding your comment, the authors specified in [L168 - L170] that indeed some participants had a glucose concentration higher than 85 mg/dl.
“In the particular case of the sample of 20 people, the glucose concentration of some participants was higher than 85 mg/dl.”
Reviewer 2 Report
The paper aims to create a mathematic model of glucose excursions in healthy people in response to dietary input and physical activity it is an interesting concept, and would be useful alongside similar studies in patients with diabetes. However, there are very few real patients used to test this model, in my opinion insufficient, with the majority of tests being performed with virtually constructed patients, and I am not convinced of the real-world applications for an in silico model tested principally in in silico patients. As discussed with the editors, the types of mathematical models are outside my area of expertise so I am unable to comment. The results and graphs need to be more clearly labelled as to whether they refer to real or virtual patient data.
[lines 1-6] the first two sentences of the abstract are fairly repetitive. There are many other instances of poor and confusing sentence structure throughout the manuscript.
[lines 153-155] there is a very big age spread for such a small sample size. If age is a significant confounder wouldn’t it be better to focus on one age group or increase the sample size to have sufficient numbers in each group?
Author Response
Response to Reviewer 2 Comments
Point 1: The paper aims to create a mathematical model of glucose excursions in healthy people in response to dietary input and physical activity. It is an interesting concept, and would be useful alongside similar studies in patients with diabetes. However, there are very few real patients used to test this model, in my opinion insufficient, with the majority of tests being performed with virtually constructed patients, and I am not convinced of the real-world applications for an in silico model tested principally in in silico patients. As discussed with the editors, the types of mathematical models are outside my area of expertise so I am unable to comment.
Response 1: Based on their feedback the number of the sample of individuals was expanded. Using a subsample from the data presented in the National Health and Nutrition Survey 2018 (ENSANUT 2018) 69 people between 20 to 60 years of age were added to the study. These results show that within the established framework of the project environment, we can be certain that the cases constructed virtually represent a significant approximation to real behavior, and that they allow us to verify that these measurements can be taken with a high degree of certainty.
Point 2: The results and graphs need to be more clearly labelled as to whether they refer to real or virtual patient data.
Response 2: We thank you for your comments. The authors corrected this detail in the graphical results, denoting the virtual patient population as the virtual population in Figures 8 and 10, in response to your thoughtful suggestion.
Point 3: [lines 1-6] the first two sentences of the abstract are fairly repetitive. There are many other instances of poor and confusing sentence structure throughout the manuscript.
Response 3: Concerning your comment, the authors have tried to improve the wording, trying to be clearer in each of the sentences of the manuscript.
An example of this is the correction in sentence L1-L6:
“This work presents a mathematical model of homeostasis dynamics in healthy individuals, focusing on the generation of conductive data on glucose homeostasis throughout the day under dietary and physical activity factors.”
Point 4: [lines 153-155] there is a very big age spread for such a small sample size. If age is a significant confounder, wouldn’t it be better to focus on one age group or increase the sample size to have sufficient numbers in each group?
Response 4: The authors thank you for your comment in response to it, we bring to your knowledge that based on a subsample of ENSANT 2018, we increased in number of physical persons (between 20 to 60 years old) considered. Although it is true that age is an extremely important factor in the degradation of glucose homeostasis. In the mathematical model this effect is not directly considered, but since it is not an epidemiological design, no confusion is generated by this variable.
We hope in future work to consider this effect to improve the approximations generated from in silico experimentation.
Reviewer 3 Report
Change the title : Dynamic analysis of glucose homeostasis in virtual patients, a comparison between different sedentary behaviors:
Dynamic of glucose homeostasis in virtual patients: A comparison between different behaviors
Abstract section
- The summary is one of the most important parts of your manuscript.
Well written summaries improve the impact of your paper and expedite Peer Review. Summaries should be specific to your findings, aims and conclusions.
- Example data demonstrating important comparisons.
- Concise conclusions in one or two sentences
- If authors worked on physically or sedentary patients, of physically, the title should be changed and improved
- Please improve the grammar using concise, clear sentences. You should seek the help of an English speaking colleague if necessary.
Introduction section
- You should aim to improve the focus of both the Introduction and Discussion sections upon the latest research. In what is a very active area of research too many of the articles cited are more than 5 years old and too few less than 2 years old. Greater emphasis should be made of the most recent research from authoritative international Q1
- This section is very long
- L42-44, what is the link between mathematical models and in silico studies?
- Authors showed several examples of mathematical model applied in many cases and disease, Please specify the introduction section on diabetes cases with some mathematical comparison
- L82/96, delete this sentence
Materials and Methods section
- Why authors used the 2 models: Bergman’s minimal model and combining the models proposed in Orozco-López et al.] and Roy and Parkermodels
- Can authors add a section “Statistical analysis” to correlate all results
- Resultas part
- Effect of diet and physical activity on glucose concentration and Variation of blood glucose concentration in the participant should be correlated
Discussion part
This part is weak, authors should compare these results with other works
Author Response
Response to Reviewer 3 Comments
Point 1: Change the title: Dynamic analysis of glucose homeostasis in virtual patients, a comparison between different sedentary behaviors:
Dynamic of glucose homeostasis in virtual patients: A comparison between different behaviors
Response 1: The authors have modified the title of the manuscript according to the reviewer's comment. New title: “Dynamic of glucose homeostasis in virtual patients: A comparison between different behaviors”
Point 2: Abstract section
- The summary is one of the most important parts of your manuscript.
- Well written summaries improve the impact of your paper and expedite Peer Review. Summaries should be specific to your findings, aims and conclusions.
- Example data demonstrating important comparisons.
- Concise conclusions in one or two sentences
- If authors worked on physically or sedentary patients, of physically, the title should be changed and improved
Response 2: Regarding your comment, the authors rewrote the abstract section.
Point 3: Please improve the grammar using concise, clear sentences. You should seek the help of an English speaking colleague if necessary.
Response 3: As you mentioned, it is necessary to improve the grammar of the text. In response to your comment, the authors contacted a proofreader to improve the grammar of the document.
Introduction section
- Point 4: You should aim to improve the focus of both the Introduction and Discussion sections upon the latest research. In what is a very active area of research too many of the articles cited are more than 5 years old and too few less than 2 years old. Greater emphasis should be made of the most recent research from authoritative international Q1.
Response 4: The authors reduced to a minimum the amount of literature older than 5 years used and added literature less than 2 years old throughout the document. Previously there were a total of 36 references of which 9 were older than 5 years, now there are a total of 42 references of which 5 are older than 5 years.
- Point 5: This section is very long
Response 5: Regarding their comments, the authors eliminated the suggested sentences and tried to reduce this section.
- Point 6: L42-44, what is the link between mathematical models and in silico studies?
Response 6: Mathematical models are a tool to develop mathematical computations for generating approximations of the dynamics of glucose homeostasis. According to these computations it is possible to generate in-silico experimentation. Regarding your comment the authors added the following sentence in the introduction section:
“[L36 - L39] From the mathematical point of view, the problem of metabolic diseases has been approached on the basis of in-silico experiments, which in conjunction with mathematical models generate numerical approximations of the behavior of glucose homeostasis when food intake occurs.”
- Point 7: Authors showed several examples of mathematical models applied in many cases and diseases. Please specify the introduction section on diabetes cases with some mathematical comparison.
Response 7: The authors thank you for your comment. In response to your suggestion, the authors would like to comment that the literature contains several works such as those mentioned in the introduction, where the use of mathematical models to make in silico comparisons with patients with T1 and T2 DM can be observed. This clearly shows the importance of the use in simulation of virtual patients to perform tests and experiments without exposing or risking the patients, since the models have been previously validated with experimental data. The following paragraph refers to his comment in the manuscript.
“[L66 - L72] In Alkhateeb et al. [26], studies on minimum order mathematical models for the development of VP that consider variations of the glycemic profile in patients with T1DM under food intakes as well as physical activity are presented; a simulator for the development of VP with T1DM under conditions of physical activity, insulin administration and food intakes is shown in Kartono et al. [27], Resalat et al. [28], Garcia-Tirado et al. [18] and Moser et al. [29] research concerning the effects of physical activity on glucose homeostasis.”
- Point 8: L82/96, delete this sentence
Response 8: The authors of the paper have deleted this sentence.
Materials and Methods section
- Point 9: Why authors used the 2 models: Bergman’s minimal model and combining the models proposed in Orozco-López et al.] and Roy and Parker Models.
Response 9: The authors are grateful for and consider important the commentary, for this particular case, where these two models were taken into consideration due to the characteristics that each of them contributes to the mathematical analysis. Orozco-López presents a mathematical description of glucose homeostasis in healthy people, while Roy and Parker emphasize the effect of physical activity on glucose variation. Considering that both representations are variants of Bergman's minimal model, it is possible to make such a consideration to achieve the objective of correlating dietary factors and the development of physical activity in relation to the variation of glucose homeostasis in healthy individuals. This new model allows the generation of complementary information that is not considered in each of the models independently
- Point 10: Can authors add a section “Statistical analysis” to correlate all results?
Response 10: Because our study is not based on some type of epidemiological design, we do not consider the inclusion of the statistical analysis section necessary.
In the methodology section, we mention the use of measures of central tendency to summarize the characteristics of the participants. However, each of these characteristics is used in the mathematical model that is the object of this study.
Results part
- Point 11: Effect of diet and physical activity on glucose concentration and Variation of blood glucose concentration in the participant should be correlated
Response 11: In response to your comment, the authors would like to inform you that, in order to present the correlations of the effect of dietary factors and physical activity on blood glucose variation, section 3.1 "Effect of primary and secondary factors on the behavior of glucose homeostasis" has been included. Within this section, Figure 4 is presented, which shows the pronounced increases in glucose levels derived from dietary intake, while in the case of physical activity, discrete decreases in glucose levels resulting from the development of physical activity are presented.
It should be noted that this correlation is also presented indirectly in Figure 11, where an analysis is made of the performance of the mathematical model when considering the effect of physical activity on the decrease in blood glucose levels.
Discussion part
- Point 12: This part is weak, authors should compare these results with other works
Response 12: Effectively this section had many weaknesses. According to the reviewers' comments, this entire section was rewritten. The aspects addressed were the following: 1) General description of the problem, 2) Link between mathematical models and in-silico experimentation, 3) Validation using the Monte Carlo approach, 4) General discussion of the results obtained, 5) Usefulness of the model for health specialists, and 6) Comparison between current works.
Round 2
Reviewer 2 Report
The authors have addressed all of my concerns. In my opinion, it would have been better if the additional physical participants could have had as many attribute markers fully described as in the original 20, but I understand that this data was not available and keeping the two cohorts seperately in the results has meant that this was able to be taken into account.
20-60 is still a reasonable age spread but since the authors state that it has no significant affect on their model it is acceptable.
Reviewer 3 Report
accept as it